# Neoadjuvant and adjuvant pembrolizumab in advanced high-grade serous carcinoma: the randomized phase II NeoPembrOV clinical trial

This open-label, non-comparative, 2:1 randomized, phase II trial (NCT03275506) in women with stage IIIC/IV high-grade serous carcinoma (HGSC) for whom upfront complete resection was unachievable assessed whether adding pembrolizumab (200 mg every 3 weeks) to standard-of-care carboplatin plus paclitaxel yielded a complete resection rate (CRR) of at least 50%. Postoperatively patients continued assigned treatment for a maximum of 2 years. Postoperative bevacizumab was optional. The primary endpoint was independently assessed CRR at interval debulking surgery. Secondary endpoints were Completeness of Cytoreduction Index (CCI) and peritoneal cancer index (PCI) scores, objective and best response rates, progression-free survival, overall survival, safety, postoperative morbidity, and pathological complete response. The CRR in 61 pembrolizumab-treated patients was 74% (one-sided 95% CI = 63%), exceeding the prespecified ≥50% threshold and meeting the primary objective. The CRR without pembrolizumab was 70% (one-sided 95% CI = 54%). In the remaining patients CCI scores were ≥3 in 27% of the standard-of-care group and 18% of the investigational group and CC1 in 3% of the investigational group. PCI score decreased by a mean of 9.6 in the standard-of-care group and 10.2 in the investigational group. Objective response rates were 60% and 72%, respectively, and best overall response rates were 83% and 90%, respectively. Progression-free survival was similar with the two regimens (median 20.8 versus 19.4 months in the standard-of-care versus investigational arms, respectively) but overall survival favored pembrolizumab-containing therapy (median 35.3 versus 49.8 months, respectively). The most common grade ≥3 adverse events with pembrolizumab-containing therapy were anemia during neoadjuvant therapy and infection/fever postoperatively. Pembrolizumab was discontinued prematurely because of adverse events in 23% of pembrolizumab-treated patients. Combining pembrolizumab with neoadjuvant chemotherapy is feasible for HGSC considered not completely resectable; observed activity in some subgroups justifies further evaluation to improve understanding of the role of immunotherapy in HGSC.

e-mail: isabelle.ray-coquard@lyon.unicancer.fr

In patients with stage IIIC/IV ovarian, fallopian tube, or primary peritoneal cancers, neoadjuvant chemotherapy (NACT) followed by interval debulking surgery (IDS) demonstrated non-inferior overall survival (OS) to primary debulking surgery (PDS) followed by chemotherapy in three randomized phase III trials (European Organisation for Research and Treatment of Cancer [EORTC] 55971[1,2], CHORUS[2,3], SCORPION[4,5]). Consequently, NACT is considered a valuable treatment option and is included in international guidelines[6,7] for patients in whom initial upfront complete resection is not possible or where there is a high risk of perioperative adverse effects. In France, ~60% of patients with International Federation of Gynecology and Obstetrics (FIGO) stage IIIC/IV epithelial ovarian cancer typically receive NACT before IDS[8].

Laparoscopy is often used to establish a diagnosis and evaluate the extent of tumor burden to estimate whether complete cytoreduction is feasible. The same approach is often used for IDS. Standard systemic treatment includes carboplatin plus paclitaxel every 3 weeks (q3w) for three to four cycles before IDS[6,9,10]. After IDS, three to four additional chemotherapy cycles are given, with or without bevacizumab. When the NeoPembrOV trial was designed, poly(ADP-ribose) polymerase (PARP) inhibitors were not used routinely as maintenance therapy.

The complete resection rate (CRR) after neoadjuvant carboplatin plus paclitaxel was 47–58% in the EORTC 55971 and SCORPION phase III trials[1,4]. Adding bevacizumab to NACT, as used in some circumstances and healthcare settings, did not substantially improve efficacy in two small randomized trials (focusing primarily on safety)[11,12]. Thus, there remains a need for alternative, more effective neoadjuvant strategies.

The rationale for combining NACT with checkpoint inhibitors targeting programmed cell death-1 (PD-1) or programmed death ligand-1 (PD-L1) is based on the presence of tumor-infiltrating lymphocytes in most cells expressing the targets for the immunomodulatory monoclonal antibodies, the abundance of tumor antigens available for cross-priming at the time of immunotherapy, and potential reinvigoration of T lymphocytes from the primary tumor infiltrate to tackle metastatic disease[13–16]. Additionally, administering checkpoint inhibitors before surgery provides the opportunity for in-depth mechanistic and biomarker studies, potentially opening possibilities for the development of more effective immune checkpoint inhibitor combinations[17].

The PD-1 inhibitor pembrolizumab has demonstrated significant efficacy in several tumor types (https://www.accessdata.fda.gov/drugsatfda_docs/label/2021/125514s096lbl.pdf). Phase III trials evaluating PD-L1 inhibitors in newly diagnosed and recurrent ovarian cancers have shown less encouraging results[18–21], and biomarkers to identify patients who may derive benefit are elusive. There are currently no reported results from randomized trials evaluating PD-1 inhibitors in combination with chemotherapy in ovarian cancer. Single-agent pembrolizumab demonstrated modest efficacy in heavily pretreated advanced ovarian cancer in the phase Ib KEYNOTE-028 study in patients with PD-L1-positive tumors and the phase II KEYNOTE-100 study enrolling patients irrespective of PD-L1 status[22,23]. However, higher activity was observed with pembrolizumab combined with chemotherapy in phase I/II single-arm studies in recurrent ovarian cancer[24–26].

We hypothesized that combining pembrolizumab with NACT could improve surgical outcomes (and ultimately, survival) and response rate to chemotherapy. Therefore, we designed the Neo-PembrOV trial to assess the effect of combining pembrolizumab with NACT and post-IDS therapy for advanced high-grade serous carcinoma (HGSC), and included extensive tumor and blood sample collection to explore potential markers to identify candidates for combining chemotherapy with PD-1 inhibition. Here, we report the primary results and key biomarker findings from this trial showing a high CRR with the addition of pembrolizumab to NACT and postoperative therapy for HGSC.

## Results

### Patient characteristics and IDS

Between February 26, 2018, and April 17, 2019, 91 patients were enrolled from 17 sites in France: 30 were randomized to NACT alone (standard-of-care arm) and 61 to NACT plus pembrolizumab (investigational arm). Baseline characteristics were generally well balanced between the treatment arms (Supplementary Table 1). Most patients (82%) had stage IIIC disease, and 56% had bulky disease (metastatic volume ≥5 cm). The mean peritoneal cancer index (PCI) score before IDS was 11.6 (SD 8.3) in the standard-of-care arm and 9.7 (SD 9.2) in the investigational arm. Postoperative bevacizumab was planned in all but three patients. All but four patients (one [3%] in the standard-of-care arm and three [5%] in the investigational arm) underwent IDS (Supplementary Fig. 1). The median time from randomization to IDS was 3.3 months (range 2.6–7.4 months) in the standard-of-care arm and 3.2 months (range 2.4–7.3 months) in the investigational arm.

### Efficacy

At IDS, 45 pembrolizumab-treated patients achieved Completeness of Cytoreduction Index (CCI) of 0 (CC0), giving a CRR of 74% (one-sided 95% confidence interval [CI] = 63%), meeting the predefined criterion for further clinical evaluation. In the NACT-alone arm, 21 patients (70%) achieved CC0 (one-sided 95% CI = 54%). When the four patients in each arm who received more than four neoadjuvant cycles were not included as responders (sensitivity analysis), the CRR was 67% (one-sided 95% CI = 56%) in the investigational arm and 57% (one-sided 95% CI = 40%) in the standard-of-care arm. Table 1 shows secondary response endpoints. Response Evaluation Criteria in Solid Tumours (RECIST) overall response rate (ORR) was 72% with pembrolizumab and 60% in the standard-of-care arm.

At the initial database cutoff for the primary endpoint analysis (September 30, 2020), the median duration of follow-up from the randomization date was 22 months (range 6.8–32.5 months). An updated analysis was performed to provide more mature progression-free survival (PFS) and OS results. At the data cutoff for this updated analysis (June 15, 2023), the median duration of follow-up was 52.4 months (range 24.6–62.7 months). By this date, PFS events had been recorded in 73 patients (80%) and 47 patients (52%) had died. Forty-one patients died from disease progression; in the remaining six patients, the cause of death was acute leukemia and unknown each in one patient in the standard-of-care arm, and central nervous system complications linked to infection, peritonitis after debulking surgery, unknown cause (2 years after surgery to CC0, after initiation of paclitaxel and platinum treatment at progression), and pneumopathy with acute renal failure and acute pulmonary edema (4 years after randomization and surgery), each in one patient in the investigational arm. PFS was similar in the two treatment arms (median 20.8 months [95% CI 15.0–25.7 months] in the standard-of-care arm and 19.4 months [95% CI 17.0–26.7 months] in the investigational arm; Fig. 1A). Median OS was 35.3 months (95% CI 27.1 months–not estimable) in the standard-of-care arm and 49.8 months (95% CI 36.1 months–not estimable) in the investigational arm (Fig. 1B). Three-year OS rates were 45% (95% CI 26–62%) in the standard-of-care arm and 65% (95% CI 51–76%) in the experimental arm.

### Efficacy according to *BRCA* mutation and PD-L1 status

For the exploratory subgroup analyses, *BRCA* mutation status (germline or somatic) was available for 87 patients (96% of the total population). *BRCA* mutations were detected in four patients (13%) in the NACT-alone arm and 15 (25%) in the pembrolizumab arm. No conclusions can be drawn regarding the relative efficacy in *BRCA*-mutated or wild-type *BRCA* subgroups given the small patient numbers, but there

**Table 1 | Summary of response results**

| Endpoint | NACT alone (n = 30) | NACT + pembrolizumab (n = 61) |
|---|---|---|
| IDS performed | 29 (97) | 58 (95) |
| Primary endpoint: CRR [one-sided 95% CI] | 21 (70) [54–] | 45 (74) [63–] |
| Completeness of Cytoreduction Index score | | |
| CC0 (no residual disease) | 21 (70) | 45 (74) |
| CC1 (≤0.25 cm residual disease) | 0 | 2 (3) |
| CC ≥3 or biopsy only | 8 (27) | 11 (18) |
| Mean change in PCI score between baseline and IDS (SD) | (n = 26) –9.6 (8.6) | (n = 54) –10.2 (9.3) |
| RECIST response after NACT [95% CI]ᵃ | 18 (60) [41–77] | 44 (72) [59–83] |
| Complete/partial response | 2 (7)/16 (53) | 2 (3)/42 (69) |
| Stable disease | 11 (37) | 14 (23) |
| Progression | 0 | 2 (3) |
| Best overall responseᵃ,ᵇ | 25 (83) | 55 (90) |
| Complete response including CC0 at IDS | 22 (73) | 45 (74) |
| Partial response | 3 (10) | 10 (16) |
| Stable disease | 4 (13) | 5 (8) |

Data are no. (%) unless otherwise noted.

*CC* completeness of cytoreduction, *CI* confidence interval, *CRR* complete resection rate, *IDS* interval debulking surgery, *NACT* neoadjuvant chemotherapy, *PCI* peritoneal cancer index, *RECIST* Response Evaluation Criteria in Solid Tumours, *SD* standard deviation.

ᵃNot evaluable in one patient in each arm.

ᵇDefined as the best response observed at any time from the date of randomization until the end of treatment.

was no clear association between *BRCA* mutation status and efficacy of pembrolizumab (Fig. 2). PD-L1 status was available for 85 patients (93%). In exploratory subgroup analyses, there was a suggestion of enhanced PFS and OS with pembrolizumab in patients with a combined positive score (CPS) ≥10 (Fig. 3).

### Treatment exposure and safety

Treatment compliance at the time of the primary analysis (data cutoff September 30, 2020), is shown in Tables 2, 3. Overall, there was no excess of chemotherapy dose reductions or premature chemotherapy discontinuation in the pembrolizumab arm during NACT or after IDS.

During NACT, grade ≥3 adverse events (AEs) were less common with pembrolizumab-containing therapy than NACT alone (Table 4). The most common grade ≥3 AE was neutropenia (13% in both arms). Investigators considered grade ≥3 AEs to be related to pembrolizumab in two patients (3%) receiving pembrolizumab-containing therapy (anemia in one patient, cachexia in one patient). Additional grade 3/4 AEs with pembrolizumab occurring during NACT included cerebral hemorrhage, pulmonary embolism, venous thrombosis, brain empyema, and asthenia (Table 5). One patient receiving pembrolizumab died from AEs (hemorrhagic shock, staphylococcal sepsis, febrile bone marrow aplasia, and cerebrovascular accident, all recorded as grade 5 but none considered related to treatment by the investigator or sponsor). All-grade AEs were typical of NACT, the most common in both arms being asthenia, nausea, anemia, and alopecia. All-grade AEs with >5% higher incidence in patients treated with pembrolizumab versus NACT alone were asthenia, nausea, abdominal pain, neutropenia, rash, hypertension, myalgia, paresthesia, hypothyroidism, hyperthyroidism, and constipation (Tables 4, 6).

Postoperative complications occurred in four patients (13%) in the NACT-alone arm and 13 patients (21%) receiving pembrolizumab with NACT, including one death from postoperative peritonitis (Supplementary Table 2). The most common postoperative complication in the investigational arm was infection/post-surgical fever (n = 4).

Across the entire treatment period, grade ≥3 AEs were recorded in 20 patients (67%) receiving NACT alone and 46 (75%) receiving pembrolizumab plus NACT (Supplementary Table 3). In the pembrolizumab arm, grade ≥3 AEs were considered related to pembrolizumab in 18% of patients (two patients with anemia [with thrombocytopenia in one patient], one patient with diarrhea/

dehydration/hypokalemia, one patient with ALT increased/AST increased/blood alkaline phosphatase increased/GGT increased, and one patient each with congestive cardiomyopathy, hypothyroidism, aptyalism, autoimmune hepatitis, cholestasis, cachexia, and acute kidney injury). There were two grade 5 AEs (one in each arm, already described above). Pembrolizumab was discontinued prematurely because of AEs in 14 patients (23%; details in Table 3) after a median of 12.5 months (range 0.7–20.2 months).

### Discussion

This trial evaluating the PD-1 inhibitor pembrolizumab for HGSC met its predefined primary objective, demonstrating a CRR of ≥50% with the addition of pembrolizumab to NACT and adjuvant chemotherapy, with or without postoperative bevacizumab, in patients with advanced HGSC not considered to be optimally resectable. However, the CRR in the standard-of-care group was higher than the expected 50% CRR with NACT alone, perhaps partially explained by the exclusion of patients anticipated to have residual disease after IDS. Furthermore, CRRs in both treatment groups were reduced in a sensitivity analysis adjusting for patients receiving >4 neoadjuvant cycles (CRR of 67% in the investigational arm and 57% in the standard-of-care arm), suggesting that the additional neoadjuvant cycles may have contributed to the higher-than-expected CRRs in both arms. On the other hand, a recent analysis from the French Epidemiological Strategy and Medical Economics (ESME) cohort suggested that although delayed IDS (after ≥5 neoadjuvant cycles) is very common in routine practice in France, such a delay is associated with worse PFS than standard IDS (after three to four neoadjuvant cycles)[27].

In NeoPembrOV, the RECIST ORR was 72% with pembrolizumab-containing therapy and 60% with chemotherapy alone. PFS was overlapping in the two treatment arms, but there was a suggestion of an emerging effect on OS. The addition of pembrolizumab was not associated with increased toxicity during NACT, nor concerning effects on postoperative safety, although 23% of patients discontinued pembrolizumab prematurely because of AEs.

The 74% CRR compares favorably with CRRs reported for chemotherapy alone in the EORTC 55971[1] and SCORPION phase III trials[4] and with results from randomized phase II trials evaluating neoadjuvant bevacizumab-containing regimens[11,12]. In a non-comparative French study, the CRR with bevacizumab plus NACT was 59%[11]. In a

A

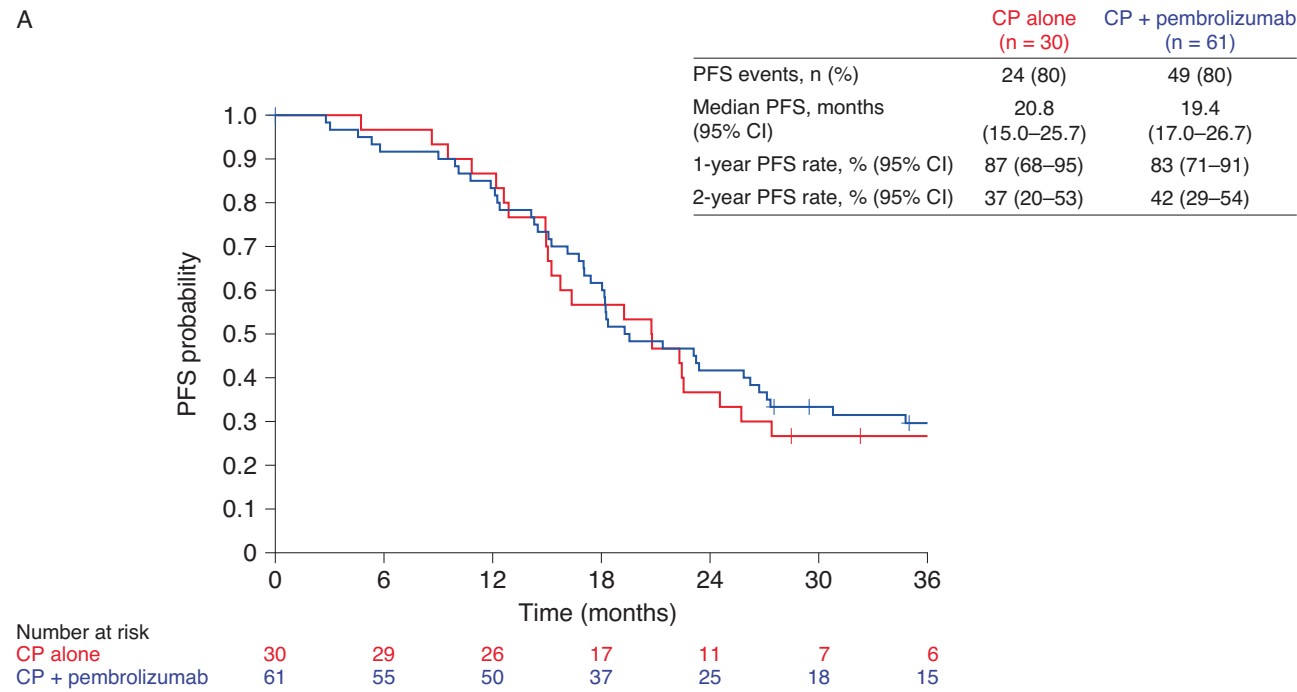

| | CP alone (n = 30) | CP + pembrolizumab (n = 61) |
|---|---|---|
| PFS events, n (%) | 24 (80) | 49 (80) |
| Median PFS, months (95% CI) | 20.8 (15.0–25.7) | 19.4 (17.0–26.7) |
| 1-year PFS rate, % (95% CI) | 87 (68–95) | 83 (71–91) |
| 2-year PFS rate, % (95% CI) | 37 (20–53) | 42 (29–54) |

Number at risk

| | | | | | | | |
|---|---|---|---|---|---|---|---|
| CP alone | 30 | 29 | 26 | 17 | 11 | 7 | 6 |
| CP + pembrolizumab | 61 | 55 | 50 | 37 | 25 | 18 | 15 |

B

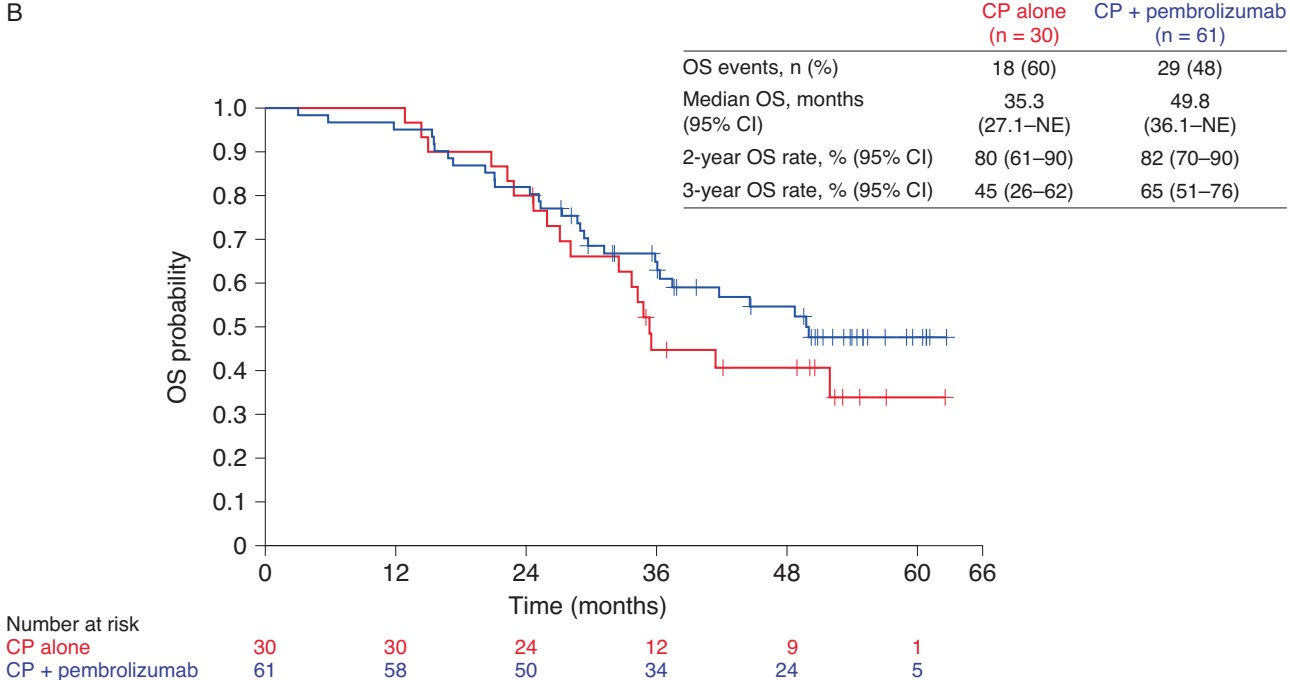

| | CP alone (n = 30) | CP + pembrolizumab (n = 61) |
|---|---|---|
| OS events, n (%) | 18 (60) | 29 (48) |
| Median OS, months (95% CI) | 35.3 (27.1–NE) | 49.8 (36.1–NE) |
| 2-year OS rate, % (95% CI) | 80 (61–90) | 82 (70–90) |
| 3-year OS rate, % (95% CI) | 45 (26–62) | 65 (51–76) |

Number at risk

| | | | | | | |
|---|---|---|---|---|---|---|
| CP alone | 30 | 30 | 24 | 12 | 9 | 1 |
| CP + pembrolizumab | 61 | 58 | 50 | 34 | 24 | 5 |

**Fig. 1 | Efficacy. A** PFS. **B** OS. CI confidence interval, CP carboplatin + paclitaxel, NE not estimable, OS overall survival, PFS progression-free survival. Source data are provided as a Source Data file.

Spanish randomized phase II trial, the CRR (defined as PCI = 0) was 29%, and the rate of complete resection or cytoreduction to ≤1 cm residual disease was 66%[12]. However, given the higher-than-expected CRR in the standard-of-care arm, which also compares favorably with previous trials, it seems that the inclusion criteria of the present trial may have led to a better-prognosis population than enrolled in historical trials. In addition, the trial was conducted in specialist centers with expertise in ovarian surgery. Thus, while the trial was not designed as a comparative trial, the similar outcome in a contemporary

population recruited under the same criteria and treated with NACT alone in the same centers as the pembrolizumab patients cannot be ignored.

In exploratory analyses, the more favorable outcome observed in the subgroup with higher (CPS ≥10) versus lower PD-L1 expression receiving pembrolizumab-containing therapy is consistent with previous findings from single-arm studies of pembrolizumab (with or without a PARP inhibitor) in recurrent ovarian cancer[23,28]. Given the caveats of small sample sizes in post hoc exploratory analyses of a non-

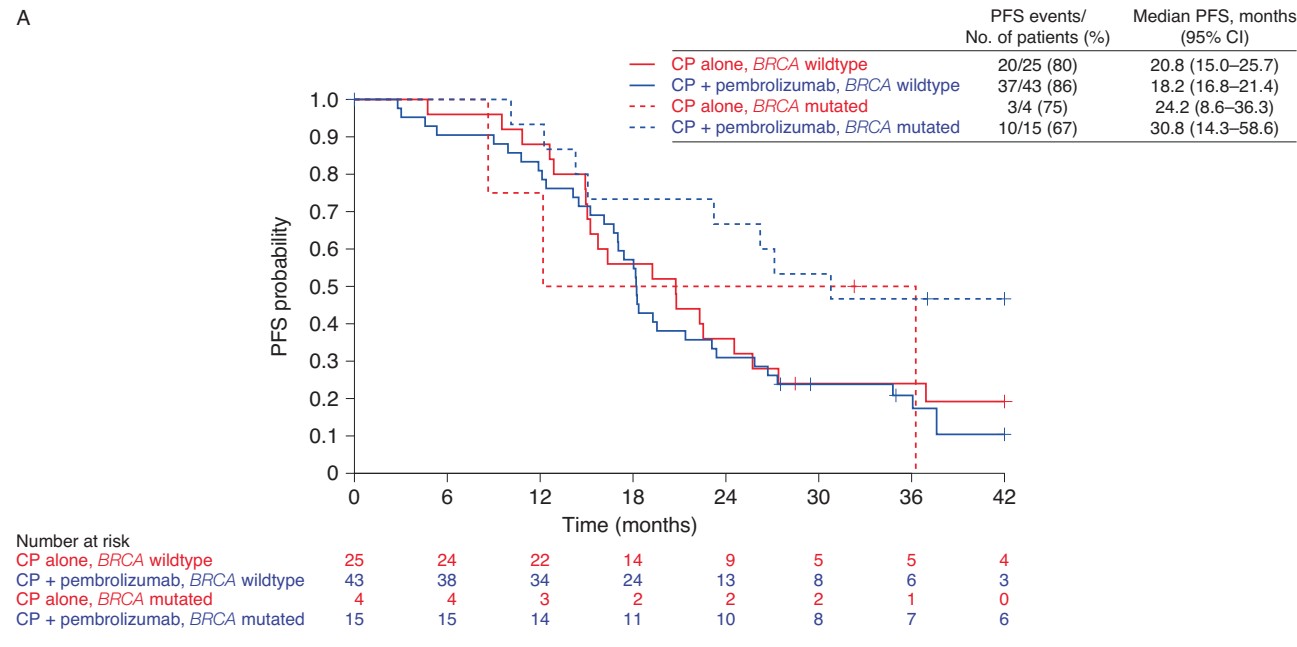

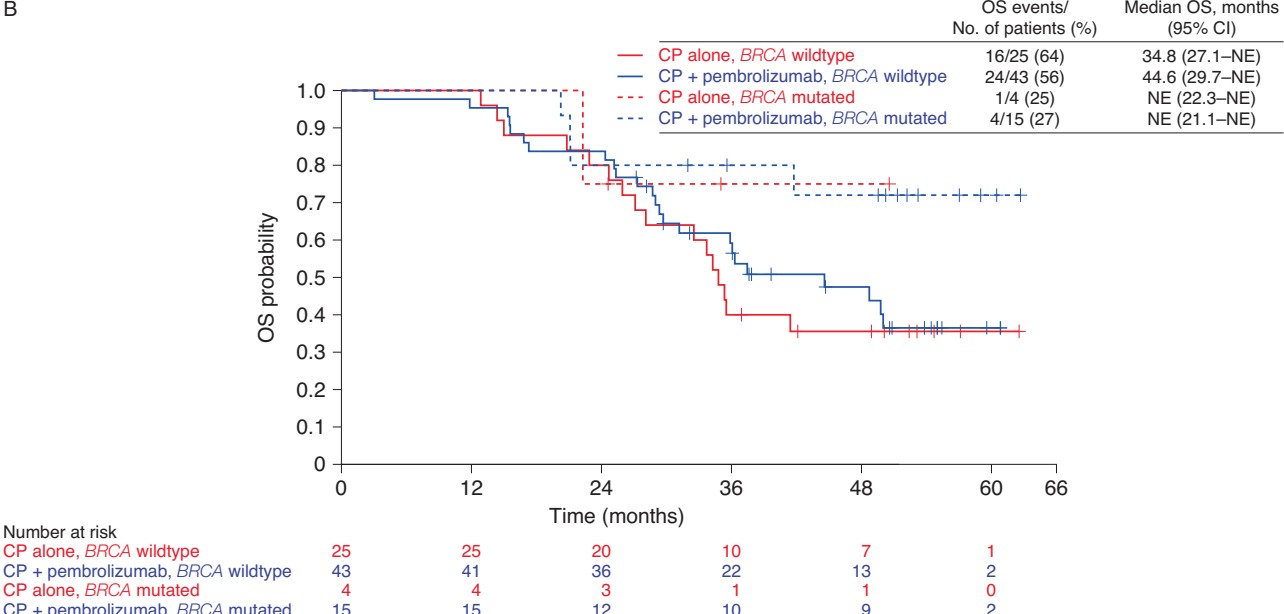

**Fig. 2 | Exploratory subgroup analyses by BRCA mutation status. A** PFS. **B** OS. CI confidence interval, CP carboplatin + paclitaxel, NE not estimable, OS overall survival, PFS progression-free survival. Source data are provided as a Source Data file.

comparative study, no conclusions can be drawn regarding the relative treatment effect of pembrolizumab according to PD-L1 over-expression. However, the modest difference in treatment effect in patients with PD-L1-positive disease makes it difficult to consider PD-L1 expression as a biomarker enabling identification of the most appro-priate candidates for immune checkpoint therapy. Furthermore, the inconclusive findings for this putative biomarker in phase III trials[19–21] are insufficient to convince clinicians and researchers of its value. The NeoPembrOV study aims to address this challenge with ongoing translational analyses of molecular biomarkers, including homologous recombination deficiency (HRD) status, as well as tumor immune contexture and angiogenesis. The translational program strives to identify the best candidates for PD-1 inhibitors in combination with NACT or, more importantly, the main mechanisms of resistance to be

targeted in combination with PD-1 inhibitors. These exploratory ana-lyses, reported in the accompanying article by Le Saux et al.[29], is important in better defining the potential contribution of PD-(L)1 inhibitors to the management of HGSC. Neoadjuvant trials with paired sample collection provide an ideal opportunity to improve under-standing of disease biology and the effect of treatment on the tumor microenvironment.

A strength of the NeoPembrOV trial is the central independent review of surgical outcomes by expert surgeons. This allows a more robust assessment of the quality of surgery than is often achieved in clinical trials of innovative therapies. Another strength of the study design is the inclusion of optional bevacizumab after surgery, which was adopted in almost 90% of patients. Bevacizumab is widely used after interval surgery in newly diagnosed ovarian cancer not

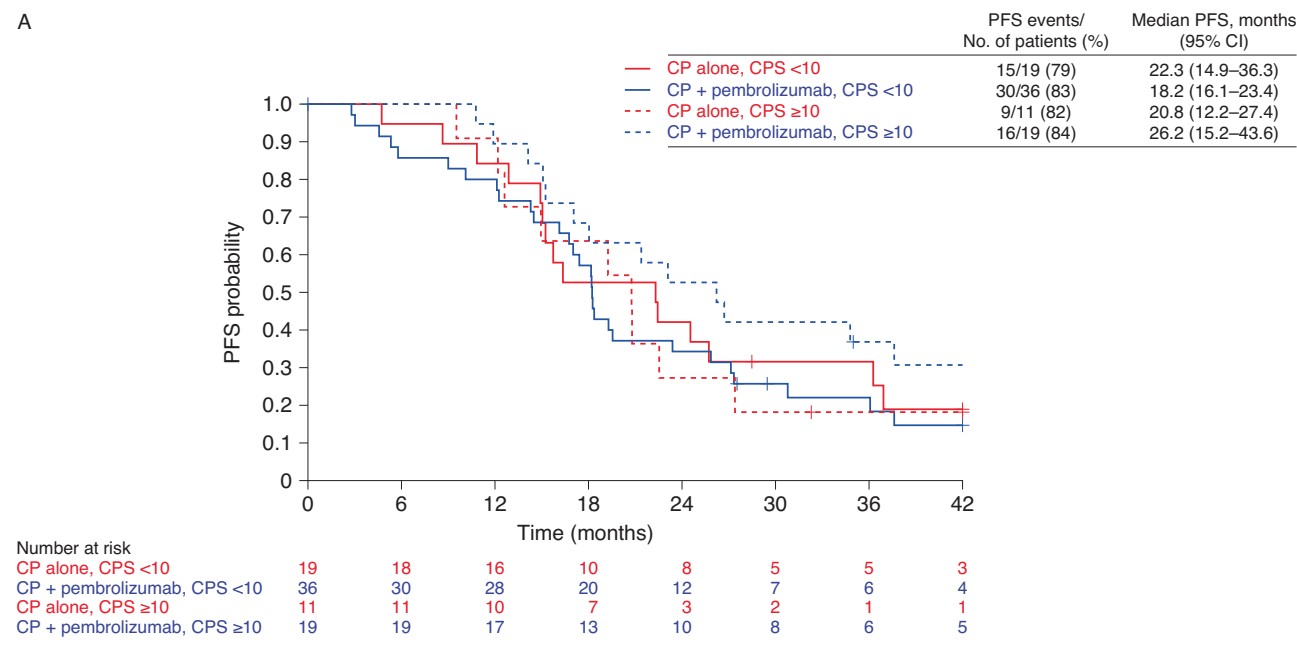

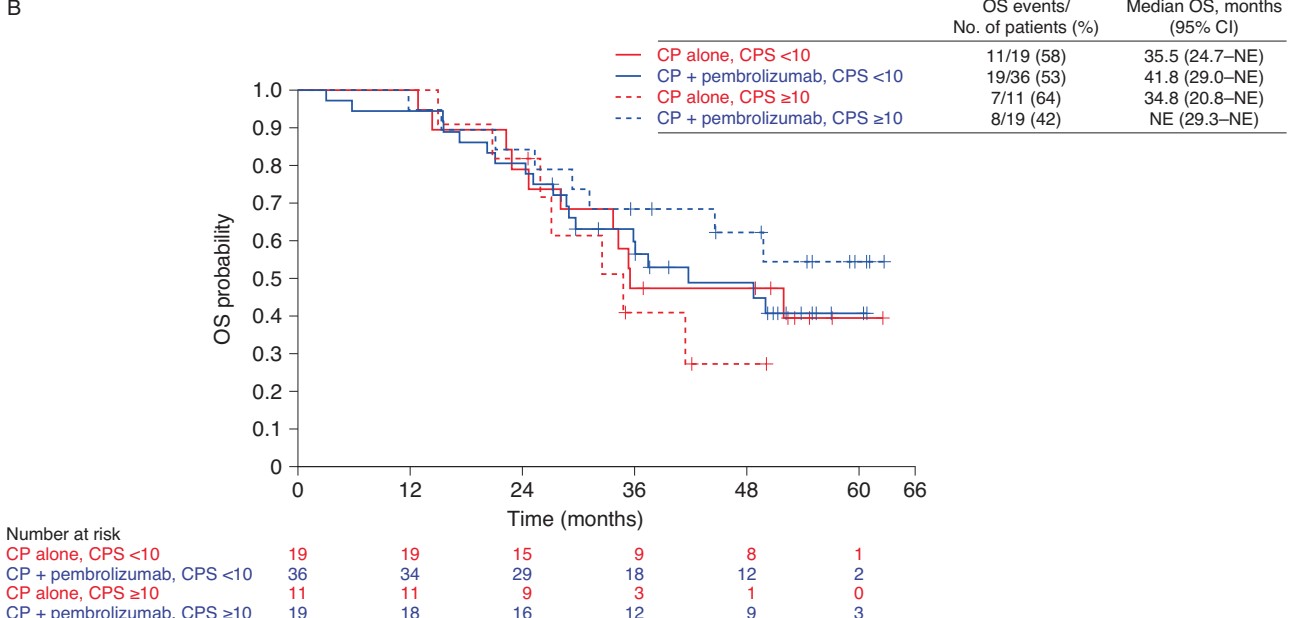

**Fig. 3 | Exploratory subgroup analyses by PD-L1 status. A** PFS. **B** OS. CI confidence interval, CP carboplatin + paclitaxel, CPS combined positive score, NE not estimable, OS overall survival, PFS progression-free survival. Source data are provided as a Source Data file.

suitable for PDS, and therefore the design reflects contemporary practice.

From a methodological perspective, NeoPembrOV raises the question of whether new therapies can be adequately assessed in randomized phase II trials using CC0 and ORR as primary endpoints to predict PFS (or efficacy more generally) in the first-line setting. The considerable efficacy in the NACT-alone standard-of-care arm (70% CRR, best overall response of 83%) may set too high a hurdle to identify new innovative therapies in this setting in small trials. There remains a need for new endpoints, such as KELIM[30,31] or biological endpoints specific to the target. Alternatively, results from the randomized phase II CHIVA trial suggest that the combination of RECIST response after neoadjuvant therapy and CC0 at debulking surgery provides a more

reliable prediction of PFS and OS than either endpoint alone. The authors concluded that a reasonable endpoint for future neoadjuvant trials could be the proportion of patients achieving both a RECIST response and CC0 at IDS[32]. Finally, the potential role of combining both bevacizumab and pembrolizumab with NACT may be of interest. Translational research reported in the accompanying article suggests that vascular endothelial growth factor 2 and/or regulatory T cells could be targeted to overcome immunoresistance to traditional immune checkpoint inhibitors in ovarian cancer[29], supporting future evaluation of bevacizumab combined with NACT before IDS.

The main limitation of the trial is the small sample size and the non-comparative statistical design, which does not allow us to estimate the contribution of pembrolizumab (including its impact on

**Table 2 | Treatment exposure (neoadjuvant phase)**

| Exposure | | CP alone (n = 30) | CP + pembrolizumab (n = 61) |
|---|---|---|---|
| Median no. of cycles (range) | Carboplatin | 4 (2–6) | 4 (1–8) |
| | Paclitaxel | 4 (1–6) | 4 (1–8) |
| | Pembrolizumab | NA | 4 (1–8) |
| Dose reduction or omission for adverse event, n (%) | Carboplatin | 2 (7) | 2 (3) |
| | Paclitaxel | 6 (20) | 3 (5) |
| | Pembrolizumab | NA | 0 |
| Early discontinuation (<4 cycles) for toxicity, progression, or death, n (%) | Carboplatin | 1 (3) | 1 (2) |
| | Paclitaxel | 2 (7) | 1 (2) |
| | Pembrolizumab | NA | 2 (3) |

*CP* carboplatin + paclitaxel, *NA* not applicable.

**Table 3 | Treatment exposure (adjuvant phase)**

| Exposure | | CP alone (n = 30) | CP + pembrolizumab (n = 61) |
|---|---|---|---|
| Median no. of cycles (range) | Carboplatin | 3 (0–6) | 3 (0–5) |
| | Paclitaxel | 3 (0–6) | 3 (0–5) |
| | Pembrolizumab | NA | 15 (0–24) |
| | Bevacizumab | 14.5 (0–22) | 15 (0–24) |
| Early discontinuation, n (%)[a] | Carboplatin | 3 (10) | 8 (13) |
| | Progression | 2 (7) | 4 (7) |
| | AE | 1 (3) | 1 (2) |
| | Other | 0 | 3 (5)[b] |
| | Paclitaxel | 6 (20) | 11 (18) |
| | Progression | 3 (10) | 4 (7) |
| | AE | 3 (10) | 4 (7) |
| | Other | 0 | 3 (5)[b] |
| | Pembrolizumab | NA | 40 (66) |
| | Progression | – | 24 (39) |
| | Toxicity | – | 14 (23)[c] |
| | Other | – | 2 (3)[d] |
| | Bevacizumab | 21 (70) | 41 (67) |
| | Progression | 15 (50) | 23 (38) |
| | Toxicity | 4 (13) | 10 (16) |
| | Other | 2 (7)[e] | 8 (11)[f] |

*AE* adverse event, *CP* carboplatin + paclitaxel, *HIPEC* hyperthermic intraperitoneal chemotherapy, *IDS* interval debulking surgery, *NA* not applicable.
[a]Before planned end of treatment per protocol.
[b]Death (n = 1), HIPEC during IDS (n = 1), investigator decision (n = 1).
[c]One case each of grade 2 colitis, grade 3 acute renal failure, grade 3 dilated cardiomyopathy, grade 3 arthralgia, grade 2 hypothyroidism, grade 1 salivary hyposecretion, grade 3 autoimmune hepatitis, grade 3 cytolysis, grade 4 sepsis, grade 1 hypereosinophilia, grade 2 pancolitis, grade 4 left ventricular ejection fraction decreased, grade 2 hepatic cytolysis, grade 2 cholangitis.
[d]Death (n = 1), HIPEC during IDS (n = 1).
[e]Breast cancer (n = 1), missing (n = 1).
[f]No bevacizumab administered (n = 5), patient died before adjuvant therapy (n = 1), patient decision (n = 1), missing (n = 1).

survival) to the treatment regimen. Plausibly, continuation of pembrolizumab in the combination arm may have been prioritized over administration of adjuvant bevacizumab (not given in >10% of patients), which could have led to bias compared with the standard-of-care arm. There were also differences in chemotherapy exposure, with 20% of patients in the standard-of-care arm versus 5% in the investigational arm having a paclitaxel dose reduction or dose omission for AEs. Furthermore, numerical imbalances in *BRCA* mutation status and PD-L1 status were apparent, and the lack of stratification for these known prognostic factors may be considered a weakness. *BRCA*-mutated tumors are associated with a better prognosis and could have influenced outcomes in the two treatment groups. Of note, the trial completed accrual before PARP inhibitors became routinely used alone or in combination with bevacizumab in the first-line maintenance setting, specifically for patients with *BRCA*-mutated and HRD tumors[33,34]. The high CRR in the standard-of-care arm (70% instead of the assumed 50%) suggests that the study population had better than expected outcomes, perhaps partially explained by the continuation of NACT beyond four cycles in some patients and surgery performed in GINECO expert centers. Finally, the choice of assay to assess PD-L1 status in patients receiving pembrolizumab can be viewed as a weakness. Many of the pivotal clinical trials of pembrolizumab have used 22C3 scored using the CPS algorithm, yet in the present study, PD-L1 was assessed using SP263. There is some evidence from non-small-cell lung cancer that these two assays are not interchangeable[35].

Despite the relatively small sample size, this randomized phase II trial demonstrates that the integration of pembrolizumab into neoadjuvant therapy is feasible and promising, at least for some HGSCs. Deeper translational research (already envisaged when the trial was designed, including systematic collection of tumor biopsies and blood before and after NACT) will be important in interpreting the impact of the investigational regimen. This may allow us to identify possible candidates and understand the target immune cell population to optimize the potential of immunotherapy in epithelial ovarian cancer. The results add to the large translational program combining spatial and molecular analyses linked to the randomized clinical trial. Tumor samples collected both before and after investigational therapy provide important new information in some subgroups. For the past 5 years, research has aimed to identify potential benefits from immune therapy in the management of high-grade ovarian cancer in large phase III trials outside large translational programs. The present randomized phase II trial provides the opportunity to explore the clinical effect of pembrolizumab added to chemotherapy and also, through extensive translational research[29], to begin to understand which populations and which combinations should be evaluated in future clinical trials. This approach may be more efficient than larger trials

without extensive comprehensive translational research to shape ongoing evaluation of immunotherapy in ovarian cancer.

## Methods
### Ethics and regulatory requirements
The study was performed in accordance with the ethical principles of the Declaration of Helsinki, the International Conference on Harmonisation/Good Clinical Practice guidelines, and the Public Health Code in France. This French trial received a favorable opinion from a French national ethics committee Comité de Protection des Personnes (CPP) Nord Ouest II, based in Amiens.

### Study design and eligibility
NeoPembrOV (Clinicaltrials.gov number NCT03275506 was an open-label randomized, non-comparative, phase II trial in women with advanced HGSC unsuitable for PDS. The protocol is available in Supplementary Note 1 in the Supplementary Materials. Eligible women had newly diagnosed (by laparoscopy or laparotomy) histologically confirmed FIGO stage IIIC or IV epithelial ovarian, fallopian tube, or primary peritoneal carcinoma that was high-grade serous or endometroid. All other histologies were excluded. Patients had to be considered unsuitable for PDS (PDS denied after evaluation by laparoscopy or laparotomy) and planned for NACT followed by cytoreductive IDS aiming for no residual disease. At inclusion, patients had to have a PCI score <30[36]. Additional inclusion criteria included age >18 and ≤75 years; Eastern Cooperative Oncology Group performance

**Table 4 | Most common adverse events during neoadjuvant therapy (any grade in ≥10% of patients) irrespective of relationship to treatment**

| No. of patients with adverse event (%) | NACT alone (n = 30) | | NACT + pembrolizumab (n = 61) | |
|---|---|---|---|---|
| | Any grade | Grade ≥3 | Any grade | Grade ≥3 |
| Any | 29 (97) | 14 (47) | 61 (100) | 22 (36) |
| Asthenia | 13 (43) | 0 | 35 (57) | 2 (3) |
| Nausea | 9 (30) | 0 | 26 (43) | 0 |
| Anemia | 11 (37) | 2 (7) | 20 (33) | 3 (5) |
| Alopecia | 8 (27) | 1 (3) | 18 (30) | 1 (2) |
| Abdominal pain | 6 (20) | 1 (3) | 16 (26) | 1 (2) |
| Constipation | 5 (17) | 0 | 15 (25) | 0 |
| Peripheral neuropathy | 7 (23) | 2 (7) | 14 (23) | 0 |
| Arthralgia | 6 (20) | 0 | 13 (21) | 0 |
| Neutropenia | 4 (13) | 4 (13) | 13 (21) | 8 (13) |
| Diarrhea | 5 (17) | 0 | 10 (16) | 0 |
| Rash | 1 (3) | 0 | 10 (16) | 0 |
| Thrombocytopenia | 5 (17) | 0 | 7 (11) | 2 (3) |
| Hypertension | 1 (3) | 0 | 6 (10) | 2 (3) |
| Decreased appetite | 2 (7) | 0 | 6 (10) | 1 (2) |
| Myalgia | 0 | 0 | 6 (10) | 0 |
| Leukopenia | 5 (17) | 0 | 3 (5) | 1 (2) |
| Fatigue | 3 (10) | 0 | 3 (5) | 0 |

status ≤2; enrollment (informed consent) ≤8 weeks after diagnosis; and adequate hepatic, renal, and bone marrow function. All patients had to provide blood samples and tissue from a newly obtained (<8 weeks before starting study treatment) core or excisional biopsy of a tumor lesion. PD-L1 expression was evaluated retrospectively by immunohistochemistry on both tumor cells and immune cells using the Ventana PD-L1 (SP263) assay (Roche Diagnostics, Zug, Switzerland), scored using CPS algorithms. *BRCA1/2* mutation status was tested locally according to standard practice. HRD status was assessed using the validated ShallowHRDv2 assay[37]. All patients provided written informed consent before undergoing any study-specific procedures.

### Treatment and study procedures

Eligible patients recruited by participating investigators were stratified by center, FIGO stage (IIIC versus IV), metastasis volume (<5 cm versus ≥5 cm), and planned bevacizumab use after IDS (yes versus no). Metastatic volume was included as a stratification factor based on the slightly worse OS observed with NACT and IDS compared with PDS in the small subgroup of patients with metastatic volume <5 cm in the EORTC 55971 trial[1]. Patients were randomized in a 1:2 ratio using a web-based system (Euraxi Pharma, https://ecrf.euraxipharma.fr/CSOnline/) and a minimization procedure to receive chemotherapy alone (standard-of-care arm) or combined with pembrolizumab (investigational arm). All patients were to receive four cycles of chemotherapy (carboplatin area under the curve 5 or 6 plus paclitaxel 175 mg/m² q3w) before IDS, followed by two to five cycles of the same chemotherapy doublet after IDS. Patients randomized to the investigational arm received pembrolizumab 200 mg q3w during chemotherapy and as maintenance therapy for up to 2 years in total (15 months after surgery). Dose modifications for toxicity are described in the protocol. Treatment was continued until unacceptable toxicity or intercurrent illness preventing further treatment, severe non-compliance, patient or investigator decision, or objective radiologic disease progression according to RECIST version 1.1, unless the investigator considered the patient still to be benefiting from treatment and other discontinuation

**Table 5 | Less common adverse events during NACT (any grade ≥3 with pembrolizumab irrespective of relationship to treatment not shown in Table 4)**

| No. of patients with grade ≥3 adverse event (%) | NACT alone (n = 30) | NACT + pembrolizumab (n = 61) |
|---|---|---|
| Pulmonary embolism | 1 (3) | 2 (3) |
| Intestinal obstruction | 1 (3) | 1 (2) |
| Pyelonephritis | 1 (3) | 1 (2) |
| Sepsis | 1 (3) | 1 (2)[a] |
| Brain empyema | 0 | 1 (2)[a] |
| Perinephric abscess | 0 | 1 (2)[b] |
| Peritonitis | 0 | 1 (2)[b] |
| Staphylococcal sepsis | 0 | 1 (2)[b] |
| Central bone marrow aplasia | 0 | 1 (2) |
| Leukocytosis | 0 | 1 (2) |
| Intra-abdominal fluid collection | 0 | 1 (2) |
| C-reactive protein increased | 0 | 1 (2) |
| Cachexia | 0 | 1 (2) |
| Cerebral hemorrhage | 0 | 1 (2) |
| Cerebral thromboembolic event | 0 | 1 (2) |
| Cough | 0 | 1 (2) |
| Venous thrombosis | 0 | 1 (2) |
| Hemorrhagic shock | 0 | 1 (2) |

Grade ≥3 events in the NACT-alone arm each in one patient (3%): vomiting, adverse drug reaction, blood alkaline phosphatase increased, central neurotoxicity, dyspnea, dermatitis bullous, pruritus, and stoma closure.
[a]Both in the same patient.
[b]All in the same patient.

**Table 6 | Additional adverse events of special interest during neoadjuvant therapy irrespective of relationship to treatment**

| No. of patients with adverse event (%) | NACT alone (n = 30) | | NACT + pembrolizumab (n = 61) | |
|---|---|---|---|---|
| | Any grade | Grade ≥3 | Any grade | Grade ≥3 |
| Hypothyroidism | 0 | 0 | 5 (8) | 0 |
| Hyperthyroidism | 0 | 0 | 4 (7) | 0 |
| Thyroid disorder | 0 | 0 | 1 (2) | 0 |
| Thyroiditis | 0 | 0 | 1 (2) | 0 |
| Abdominal pain upper | 1 (3) | 0 | 2 (3) | 0 |
| Dyspepsia | 0 | 0 | 1 (2) | 0 |
| Mucosal inflammation | 0 | 0 | 3 (5) | 0 |
| Hepatocellular injury | 1 (3) | 0 | 1 (2) | 0 |
| Infusion-related reaction | 0 | 0 | 1 (2) | 0 |
| Thyroid function test abnormal | 0 | 0 | 1 (2) | 0 |
| Interstitial lung disease | 0 | 0 | 2 (3) | 0 |

criteria were not met. After IDS, bevacizumab 15 mg/kg q3w was permitted for 15 months in total in both treatment arms at the investigator's discretion.

Tumors were assessed according to RECIST version 1.1 by cross-sectional imaging at baseline, within 7 days before cycle 3, at the end of NACT, at the end of adjuvant therapy, every 6 months during the first year of maintenance therapy, and then at the time of suspected disease progression. Four weeks after the fourth cycle of NACT, patients

underwent a mandatory laparoscopy with PCI assessment and resectability evaluation with the aim of performing IDS. AEs were assessed at every treatment cycle according to the National Cancer Institute Common Terminology Criteria for Adverse Events version 4.03. An independent data monitoring committee reviewed safety data at regular intervals throughout the study.

## Outcome measures

The study was preregistered on September 7, 2017. Originally the primary endpoint was CRR after IDS, defined as the removal of all macroscopic residual tumor (CCI = 0; CC0) as assessed by the investigator. This was modified in a protocol amendment on June 3, 2020, to CRR at IDS as assessed by a blinded independent centralized review by two surgical experts and the coordinating investigator, who reviewed the anonymized operative and pathologic reports of all patients at screening, at IDS, and at other debulking surgery.

Secondary efficacy endpoints included CCI score by local assessment, PCI score by local and central assessment (added at the June 3, 2020 protocol amendment, to be reported separately), ORR after four neoadjuvant cycles according to RECIST version 1.1, best overall response to the global strategy, PFS according to RECIST version 1.1 (defined as the interval between randomization and date of disease progression or death, whichever occurred first), and OS. Other secondary endpoints included safety during NACT and in the adjuvant setting, postoperative mortality, postoperative morbidity according to modified Clavien-Dindo classification, and pathological complete response (pCR). pCR is not reported as it required a complete examination of the epiploon surgical part, which was not performed by all centers.

## Statistics

Assuming a 50% CRR at IDS with NACT alone, as reported in the literature[1,4], the planned sample size of 90 patients (60 in the investigational arm, 30 in the standard-of-care arm) was calculated based on the A'Hern single-stage design[38], with a ≥70% success rate (CC0 at IDS) in the pembrolizumab arm considered sufficient to justify further evaluation and a <50% rate considered insufficient. The trial was designed with 90% power at a one-sided alpha of 0.05 based on CC0 in 33 of 54 evaluable patients in the pembrolizumab arm, assuming that 10% of patients would be non-evaluable. The standard-of-care arm was included to avoid selection bias, but no formal statistical testing was planned. All efficacy endpoints were evaluated in the intention-to-treat population. Safety was analyzed in all patients who received at least one dose of systemic therapy.

The secondary endpoints of PFS and OS were analyzed using the Kaplan–Meier methodology. In post hoc exploratory subgroup analyses, PFS according to PD-L1 status and *BRCA* mutation status was estimated using Kaplan–Meier methodology.

Data were collected in academic centers via an electronic case report form (CS Online Ennov Clinical version v8.2.50 powered by Euraxi Pharma, a French contract research organization). The data were monitored through on-site monitoring visits by clinical research associates according to a prespecified monitoring plan. All data were centralized in a database that was handled and controlled according to a specific data management plan, and analyzed using SAS version 9.4.

## Reporting summary

Further information on research design is available in the Nature Portfolio Reporting Summary linked to this article.

## Data availability

Data sharing in a public repository was not planned at the beginning of the study. Requests to access the deidentified data for further scientific use sent to ARCAGY-GINECO (Sébastien Armanet sarmanet@arca-gy.org) will be considered on a case-by-case basis in a timely manner beginning 3 months and ending 5 years after this article publication. Requests must contain a proposal with scientific and methodologically justified objectives. A Data Transfer Agreement will be established to provide a formal framework regarding the use of the data. The deidentified data underlying the results generated in this article are provided in the Source Data files. The study protocol is provided in the Supplementary Note 1 in the Supplementary Information. Source data are provided with this paper.

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

## Acknowledgements

The authors thank all the patients and their families, the investigators and staff (L. Terec and N. Cloarec, Avignon; L. Mansi, F. Bazan and E. Kalbacher, Besançon; F. Joly, Caen; S. Dubois, X. Durando, I. Van Praagh and M-A. Mouret-Reynier, Clermont-Ferrand; L. Favier and L. Bengrine Lefevre, Dijon; T. L'Haridon and F. Priou, La Roche-sur-Yon; L. Venat, Limoges; I. Ray-Coquard, O. Tredan and O. Derbel, Lyon; C. Cornila and J. Meunier, Orleans; F. Selle, Paris; A-M. Savoye, Reims; M. Leheurteur, Rouen; D. Berton, Saint-Herblain; O. Collard, Saint-Priest-en-Jarez; A. Martinez, R. Despax and M. Martinez, Toulouse), the ARCAGY-GINECO study team (B. Votan, S. Armanet, C. Montoto-Grillot and M. Andriamamonjy), A. Degnieau from the ARCAGY-GINECO Biobank, the biostatisticians (S. Chabaud and C. Schiffer), the pathologists (I. Treilleux, G. Bataillon and P.A. Just), the technicians (L. Odeyer and A. Vermorel) who performed the PD-L1 staining, LYRICAN for funding the PD-L1 analysis (Grant SIRIC LYriCAN INCa-DGOS-Inserm_12563), and the surgical reviewers (G. Ferron and J-M. Classe). Merck and Co. Inc. provided financial support but had no role in the design and conduct of the study; collection, management, analysis, or interpretation of the data; preparation, review, or approval of the manuscript; or decision to submit the manuscript for publication. The sponsor (ARCAGY-GINECO) was involved in the study design, data collection, and analysis and funded medical writing for this article, provided by Jennifer Kelly, MA (Medi-Kelsey Ltd, Ashbourne, UK).

## Author contributions

I.L.R.-C. had full access to all of the data in the study and takes responsibility for the integrity of the data and the accuracy of the data analysis. I.L.R.-C. conceived and designed the study. I.L.R.-C., C.S., and O.L.S. drafted the manuscript. C.S. performed the statistical analysis. I.L.R.-C. obtained funding from l'Association pour la Recherche sur le Cancer (ARC) dedicated to translational research. I.L.R.-C. supervised the study. All authors contributed to the acquisition, analysis, and interpretation of data, and reviewed, edited, and approved the manuscript for important intellectual content.

## Competing interests

I.L.R.-C. reports personal honoraria from Agenus, Blueprint, BMS, PharmaMar, Genmab, Pfizer, AstraZeneca, Roche, GSK, MSD, Deciphera, Mersana, Merck Sereno, Novartis, Amgen, MacroGenics, Tesaro, and Clovis; honoraria to her institution from GSK, MSD, Roche,

and BMS; consulting/advisory roles for AbbVie, Agenus, Advaxis, BMS, PharmaMar, Genmab, Pfizer, AstraZeneca, Roche/Genentech, GSK, MSD, Deciphera, Mersana, Merck Sereno, Novartis, Amgen, Tesaro and Clovis; research grant/funding from MSD, Roche and BMS (self) and MSD, Roche, BMS, Novartis, AstraZeneca and Merck Sereno (to institution); and travel support from Roche, AstraZeneca, and GSK. E.K. reports honoraria from AstraZeneca, Roche, Sanofi, Tesaro, and GSK; consulting/advisory roles for AstraZeneca, Roche, Sanofi, Tesaro and GSK; speakers' bureau for AstraZeneca, Roche, Sanofi, Tesaro, and GSK; and research funding from AstraZeneca, Roche, Sanofi, Tesaro and GSK. M.L. reports consulting/advisory roles for Pierre Fabre, GSK, and Daiichi; research funding from Veracyte; and travel/accommodation/expenses from Mundi Pharma, Pfizer, GSK, and MSD. L.B.L. reports travel/accommodation/expenses from Servier. F.P. reports consulting/advisory role for AstraZeneca. F.S. reports honoraria from AstraZeneca, Clovis Oncology, MSD, GSK-Tesaro, and Sandoz (Novartis); consulting/advisory roles for AstraZeneca, GSK-Tesaro and MSD; speakers' bureau for AstraZeneca, MSD, and GSK-Tesaro; and travel/accommodation/expenses from Roche, AstraZeneca, MSD, and GSK-Tesaro. E.C. reports the following, all for an immediate family member: employment with Sanofi; consulting/advisory role for MSD, BMS, Ipsen, and AstraZeneca; speakers' bureau for BMS, Ipsen, AstraZeneca, and Janssen; research funding from Pfizer; and travel/accommodation/expenses from Janssen. O.L.S. reports honoraria from MSD and Clovis, and travel/accommodation/expenses from Eisai. A.A. reports honoraria from GlaxoSmithKline. F.J. reports honoraria from Clovis, Amgen, GSK, Ipsen, BMS, MSD, AstraZeneca, Astellas, Janssen and Bayer; consultancy from AstraZeneca, GSK, Janssen and Ipsen; travel/accommodation/expenses from Ipsen and GSK; participation on a data safety monitoring/advisory board for GSK; and participation on the GINECO guidelines committee. O.T. reports honoraria from Roche, MSD/Merck, Novartis/Sandoz, Pfizer, Lilly, AstraZeneca, Daiichi Sankyo, Eisai, Pierre Fabre, Seagen, and Gilead, and research funding from Roche, MSD/Merck and BMS. The remaining authors declare no competing interests.

## Additional information

Isabelle L. Ray-Coquard [1] ✉, Aude-Marie Savoye[2], Camille Schiffler[1], Marie-Ange Mouret-Reynier [3], Olfa Derbel[4], Elsa Kalbacher[5], Marianne LeHeurteur[6], Alejandra Martinez[7], Corina Cornila[8], Mathilde Martinez[9], Leila Bengrine Lefevre[10], Frank Priou[11], Nicolas Cloarec [12], Laurence Venat[13], Frédéric Selle [14], Dominique Berton [15], Olivier Collard[16,20], Elodie Coquan[17], Olivia Le Saux [1,18], Isabelle Treilleux[1], Sophie Gouerant[6], Antoine Angelergues [14], Florence Joly[17] & Olivier Tredan[19]

[1]Groupe d'Investigateurs Nationaux pour l'Etude des Cancers Ovariens (GINECO) and Centre Léon Bérard, University Claude Bernard, Lyon, France. [2]GINECO and Institut Jean Godinot, Reims, France. [3]GINECO and Department of Medical Oncology, Centre Jean Perrin, Clermont-Ferrand, France. [4]GINECO and Institut de Cancérologie, Hôpital Privé Jean Mermoz, Lyon, France. [5]GINECO and Centre Hospitalier Universitaire Jean Minjoz, Besançon, France. [6]GINECO and Medical Oncology Department, Centre Henri-Becquerel, Rouen, France. [7]GINECO and Institut Claudius Régaud, Institut Universitaire du Cancer de Toulouse (IUCT) Oncopole, Toulouse, France. [8]GINECO and Centre Hospitalier Régional d'Orléans, Orleans, France. [9]GINECO and Clinique Pasteur, Toulouse, France. [10]GINECO and Centre Georges-François Leclerc, Dijon, France. [11]GINECO and Centre Hospitalier Départemental Vendée, La Roche-Sur-Yon, France. [12]GINECO and Centre Hospitalier Henri Duffaut d'Avignon, Avignon, France. [13]GINECO and Centre Hospitalier Universitaire Dupuytren, Limoges, France. [14]GINECO and Groupe Hospitalier Diaconesses Croix Saint-Simon, Paris, France. [15]GINECO and Institut de Cancérologie de l'Ouest, Centre René Gauducheau, Saint-Herblain, France. [16]GINECO and Institut de Cancérologie de la Loire, Saint-Priest-en-Jarez, France. [17]GINECO and Department of Medical Oncology, Centre François Baclesse, University Caen Normandie, Caen, France. [18]Cancer Research Center of Lyon (CRCL), UMR INSERM 1052, Centre Léon Bérard, CNRS 5286 Lyon, France. [19]GINECO and Department of Medical Oncology, Centre Léon Bérard, Lyon, France. [20]Present address: Center of Medical Oncology, Hôpital Privé de la Loire, Saint-Etienne, France. ✉e-mail: isabelle.ray-coquard@lyon.unicancer.fr

