## [Peer Review File · Nature Communications]

Reviewers' Comments:

Reviewer #1:

Remarks to the Author:

Thank you for the opportunity to review the NeoPembroOV study. This was a randomized phase 2 study of standard care chemotherapy alone or with pembrolizumab; bevacizumab was given at investigator choice. The study was designed with a control arm to reduce bias, but no clinical comparison between the two arms was planned.

Full credit to the authors for getting the study completed and publishing the results. My feedback is as below:

- 1) The Database cutoff was quite some time ago – September 2020. Can we expect updated data? If 21% of the patients have died as of the data cutoff, it would be helpful to have an OS Kaplan meier curve as well, at least for supplemental.
- 2) It's interesting to note that the BRCA mutations were so imbalanced across the treatment arms – 20% in the pembrolizumab arm! This is briefly acknowledged in your results section but should be emphasized given that patients with BRCA mutations have better prognosis. In fact, this factor alone may explain the improved RECIST response rate in that group.
- 3) It would be helpful to pull out which AEs were attributable to pembrolizumab in that treatment group.
- 4) Your consort diagram can likely be moved to supplemental
- 5) In figure 5, would condense all the survival curves onto a single axis to emphasize that the BRCA mutants did well regardless of treatment group, and to normalize the PFS follow up across the two groups - right now, there is 30 months of follow up for the BRCA mutants and 40 months for the wild type.

Reviewer #3:

Remarks to the Author:

The stated objective of the study by Ray-Coquard and colleagues is investigated pembrolizumab-containing neoadjuvant chemotherapy (NACT) in patients with FIGO stage IIIC/IV high-grade serous carcinoma (HGSC). This was randomized phase II study (n=91) with 2:1 randomization in favor of the experimental arm. The primary endpoint was independently assessed complete resection rate (CRR) at interval debulking surgery IDS. Of note, when the NeoPembrOV trial was designed, PARPi were not used routinely as maintenance therapy.

Overall the analyses are appropriate and while this study is not entirely novel it adds to other evidence suggesting a lack of benefit from antiPD1-chemo combinations in the frontline management of advanced stage ovarian cancer.

Critiques are included below:

1. There is concern about a number of infectious complications in the experimental arm including brain empyema. While these are separately listed in Table 3. The investigators should statistically compare and report the rates of infectious complication for the two arms, and offer a discussion of potential contributing causes.
2. In the patient with bone marrow aplasia the authors should comment on whether Hemophagocytic lymphohistiocytosis (HLH) was considered and ruled out, as this would be related to the investigational drug.
3. The authors should refrain from the biased use of phrases such as "numerically higher." It's best to avoid such ambiguous terminology any way, however when it is used only to promote the experimental arm (rather than for example when the overall PFS is numerically lower in the experimental arm), this introduces bias.
4. The proportion of BRCA mutation carriers was disproportionate in the control vs. experimental arm (10 vs. 21%) and these patients had a much better PFS. The authors should also report PFS in the two arms with BRCA mutation carriers excluded.
5. While the authors mention independent surgical review as a strength, it is unclear whether this

in fact provided any added value, because apparently the blinded centralized review was based on operative notes (by the primary surgeon) rather than a second surgeon performing an independent assessment in the OR.

Reviewer #4:

Remarks to the Author:

Abstract lines 36-40. These lines should be combined to conserve on word limit for other aspects of the abstract. Also, it is not clear from the description of the trial that the two arms are not being formally compared with this trial design. The trial is addressing the question whether carboplatin + paclitaxel + Pembrolizumab approaches has a complete resection rate of at least 70% as well as whether CCR with carboplatin + paclitaxel is consistent with historical findings.

Example: An open-label randomized phase II trial (NCT03275506) of women with FIGO stage IIIC/IV high-grade serous ovarian carcinoma (HGSC) for whom complete resection was unachievable was conducted to examine the complete resection rate (CCR) with carboplatin + paclitaxel and to assess whether adding Pembrolizumab to this combination yields a CCR of at least 70% (NCT03275506). Post-operatively patients continue assigned treatment for maximum of 2 years. BEV is optional.

Abstract line 39: Clearly state what the standard of care chemotherapy - carboplatin + paclitaxel.

Abstract line 47-48: The presentation of pre and post operative severe toxicities leave the impression that there is a difference between treatment arms in the both the pre and post operative severe toxicity rates. It is stated that there were more post-operative toxicities with PEM than pre-operatively without a formal testing. This could be quite misleading given the small sample size and not knowing at this point in the manuscript how many patients failed to initiate post operative chemotherapy or how many patients elected to receive bevacizumab. It would be more informative to provide the most common severe toxicities seen in each treatment arm and the number of patients who discontinued due to adverse events. Post surgical complications should also be presented.

Abstract line 45 states that progression-free survival is similar between the groups. When does the time clock start for PFS at time of registration? Post surgery? It is not clear how mature these data are. What is the median length of followup? How many progressions and deaths have been seen? This should be removed from the abstract because lack clarity as to what this is.

Abstract line 50. Given your study design, a comment should be made concerning both treatment approaches, not just commenting on PEM containing regimen. Was the Carbo-platin + paclitaxel similar to historical findings. Also, it is unclear why it is suggested that this be further studies in specific subgroups.

Results Line 106-107. The treatment arms should not be reference to as experimental and control arms as this trial was not designed to compare treatment arms. It would be more appropriate to refer to them as investigational and standard of care (SOC) arms or NACT+/-PEM as you do in the table and figures.

Results line 113-116 The median time from randomization to IDS was 3.3 months (range 2.6–7.4 months) in the control arm and 3.2 months (range 2.4–7.3 months) in the experimental arm. This should be replaced by the time on treatment or the median number (range) of treatment cycles administered, number of dose reductions, most common reasons for dose reductions, and the reasons for discontinuation of neo-adjuvant treatment. There should be an explanation as to why 58 of 61 NACT +PEM patients and 29/30 NACT patients went on to debulking surgery. How many patients refused to undergo post operative chemotherapy? What were the results for discontinuing post-operative chemotherapy? It is important to understand how patients fared during entire treatment course not just the results of the treatment. This leads to the suggestion that toxicity section be placed before efficacy.

Some of the items suggested to be in the text are in Table 2 but it is difficult to figure out what is going on. It is assumed that if a patient progresses (or refuses to continue on study) all treatment is discontinued so it does not make confusing to see different number of patients under each agent being taken off for disease progression (or refuses to continue on study). If the patient went off treatment for an AE, it is important to know the grade of toxicity. It is strongly suggested that a statement such as dose reductions were reported in carboplatin (X, %) and paclitaxel (Y,%) in SOC arm and in carboplatin (X, %), paclitaxel (Y,%) and pem (Z, %) in investigational arm. End of neo-adjuvant treatment reasons included: completed planned number of treatment cycle, toxicity, progression, etc. This can be repeated for post operative treatment as well.

Results Lines 123-125 should be removed. Secondary response endpoints showed numerically higher activity in pembrolizumab-treated patients, although there was no formal comparison of the two treatment arms (Table 1). The confidence intervals overlap.

Results Lines 128-130 state that the median duration of follow-up 129 was 22 months (range 6.8–32.5 months), progression-free survival (PFS) events had been recorded in 54 patients (59%), and 19 patients (21%) had died. The statistical analysis section fails to define PFS or duration of follow-up. Does the clock start at registration, at completion of surgery, or initiation of post operative chemotherapy?

Results Line 135-136. Median PFS should have accompanying 95% confidence interval.

Results 139: Efficacy according to PD-L1 and BRCA mutation status. And Figures 3-5. This entire section is an unplanned analysis not spelled out in the protocol. BRCA mutation status is not even mentioned in the protocol. There is no justification for combining treatment arms other than the small number of patients with either germline or somatic BRCA mutants. This unplanned, underpowered subset analysis of immature outcome data should be removed from the manuscript.

Discussion Lines 190-193 Unplanned sensitivity analysis should be removed as a it is unclear how this was done.

Discussion Lines 198-200 In NeoPembrOV, we observed a numerical trend in RECIST response rate favoring the addition of pembrolizumab to chemotherapy (72% versus 60% with chemotherapy alone), but no difference in PFS at the database lock for this analysis. This statement needs to be rewritten. It was clearly stated that no comparison between treatment arms were going to be made in the protocol -- there is little to no power to detect anything but a huge difference in PFS and the confidence intervals for response overlap. There is no evidence here that addition PEM is improving outcomes.

Discussion line 261: The phrase Despite the observed trend favoring pembrolizumab should be removed. It has not been demonstrated that adding pem improves outcomes.

Results Line 153-55: Overall, there was no excess of chemotherapy dose reductions or premature chemotherapy discontinuation in the pembrolizumab arm during NACT or after ID

Supplemental Table 1:

- Replace mean age and standard deviation with age categorized as 40-49, 50-59, 60-69, 70-79.
- Replace mean CA125 and standard deviation with CA125 categorized as < 500 U/mL, 500-999; 1000-1499; 1500-1999, 2000+ (as high pre-CA125 levels have been shown to be associated with suboptimal reductions)

Table 3: Clearly state in the title whether these are all toxicities regardless of attribution or at least possibility related to study meds.

Neoadjuvant and adjuvant pembrolizumab for advanced high-grade serous carcinoma (NeoPembroOV): A randomized phase II GINECO clinical trial [Ray-Coquard et al, NCOMMS-23-08443-T]

Point-by-point response to reviewers

Reviewer #1 Ovarian Cancer, clinical trials, immunotherapy (also reviewed NCOMMS-23-08444-T) (Remarks to the Author):

Thank you for the opportunity to review the NeoPembroOV study. This was a randomized phase 2 study of standard care chemotherapy alone or with pembrolizumab; bevacizumab was given at investigator choice. The study was designed with a control arm to reduce bias, but no clinical comparison between the two arms was planned.

Full credit to the authors for getting the study completed and publishing the results. My feedback is as below:

1) The Database cutoff was quite some time ago – September 2020. Can we expect updated data? If 21% of the patients have died as of the data cutoff, it would be helpful to have an OS Kaplan meier curve as well, at least for supplemental.

Response: Following this request, we undertook a datasweep including PFS and OS updates with a new data cutoff date of 15th June 2023. These results, including a KM curve of OS, are now reported in the manuscript.

2) It's interesting to note that the *BRCA* mutations were so imbalanced across the treatment arms – 20% in the pembrolizumab arm! This is briefly acknowledged in your results section but should be emphasized given that patients with *BRCA* mutations have better prognosis. In fact, this factor alone may explain the improved RECIST response rate in that group.

Response: While updating the analyses with extended follow-up, we took the opportunity to update *BRCA* status for those with unknown status at the time of the primary analysis. We have also incorporated HRD status (available from the research team, using the validated Curie HRD test [Callens C et al., ESMO Gynecology 2023] and now present this in Supplementary Table 1. With *BRCA* mutation status now available for 96% of patients, there remains a numerical imbalance between treatment arms. Importantly, however, HRD status was well balanced between the two treatment arms. We have expanded the discussion around the imbalance in *BRCA* mutation status in the discussion (lines 319–320), while taking care not to make any claims about comparisons between the two treatment arms, in line with other reviewer comments.

3) It would be helpful to pull out which AEs were attributable to pembrolizumab in that treatment group.

Response: The proportion of patients with AEs considered by the investigator to be related to pembrolizumab has been added for both the NACT phase (lines 178–180) and across the entire treatment period (lines 198–204), with the individual AEs attributed to treatment listed.

4) Your consort diagram can likely be moved to supplemental

Response: Moved as suggested

5) In figure 5, would condense all the survival curves onto a single axis to emphasize that the *BRCA* mutants did well regardless of treatment group, and to normalize the PFS follow up across the two groups - right now, there is 30 months of follow up for the *BRCA* mutants and 40 months for the wild type.

Response: The figure (now Figure 2) has been modified as requested. The median follow-up for all PFS and OS data is 52.4 months (range 24.6–62.7 months).

Reviewer #2 Ovarian Cancer, clinical trials, immunotherapy (also reviewed NCOMMS-23-08444-T) (Remarks to the Author):

The stated objective of the study by Ray-Coquard and colleagues is investigated pembrolizumab-containing neoadjuvant chemotherapy (NACT) in patients with FIGO stage IIIC/IV high-grade serous carcinoma (HGSC). This was randomized phase II study (n=91) with 2:1 randomization in favor of the experimental arm. The primary endpoint was independently assessed complete resection rate (CRR) at interval debulking surgery IDS. Of note, when the NeoPembrOV trial was designed, PARPi were not used routinely as maintenance therapy.

Overall the analyses are appropriate and while this study is not entirely novel it adds to other evidence suggesting a lack of benefit from antiPD1-chemo combinations in the frontline management of advanced stage ovarian cancer.

Critiques are included below:

1. There is concern about a number of infectious complications in the experimental arm including brain empyema. While these are separately listed in Table 3. The investigators should statistically compare and report the rates of infectious complication for the two arms, and offer a discussion of potential contributing causes.

Response: The trial was not powered or designed for statistical comparison of adverse events, and therefore any statistical comparison of infectious complications would be purely descriptive and we respectfully do not consider such a comparison to be scientifically robust. However, as some patients experienced more than one event (now detailed in footnotes to Table 3; brain empyema and sepsis occurred in one patient, and perinephric abscess occurred with peritonitis and staphylococcal sepsis in another patient), the incidence of infectious complications is quite similar in the two arms (during NACT: 2/30 patients [7%] in the standard-of-care arm and 3/61 patients [5%] in the experimental arm; overall: 3/30 patients [10%] and 5/61 patients [8%], respectively).

2. In the patient with bone marrow aplasia the authors should comment on whether Hemophagocytic lymphohistiocytosis (HLH) was considered and ruled out, as this would be related to the investigational drug.

Response: At the time of the event, the principal investigator discussed this case and it appeared to be directly related to surgical complications rather than to pembrolizumab. Following the reviewer's comment, we have re-reviewed the pharmacovigilance database and discussed this case again with the treating physician. The chronology clearly suggests a surgical complication with a postoperative abdominal abscess that was managed surgically with the addition of antibiotics and anticoagulation therapy. However, the abscess was complicated by staphylococcal septicemia complicated by a stroke and finally a hemorrhagic shock with death. We remain confident that the real issue is abscess and infection during chemotherapy delivery. Therefore personally I agree with the investigator not to consider this to be related to the systemic therapy; also adding chemotherapy to any ongoing infection can be a potential risk.

3. The authors should refrain from the biased use of phrases such as "numerically higher." It's best to avoid such ambiguous terminology any way, however when it is used only to promote the experimental arm (rather than for example when the overall PFS is numerically lower in the experimental arm), this introduces bias.

Response: Amended throughout as recommended.

4. The proportion of *BRCA* mutation carriers was disproportionate in the control vs. experimental arm (10 vs. 21%) and these patients had a much better PFS. The authors should also report PFS in the two arms with *BRCA* mutation carriers excluded.

Response: Figure 5A (now 2A) shows PFS in the *BRCA*-wildtype subgroup. While updating the analyses with extended follow-up, we took the opportunity to update *BRCA* status for

those with unknown status at the time of the primary analysis and also collected HRD status. With *BRCA* status now available for 96% of patients, there remains an apparent numerical imbalance in *BRCA* mutation status between treatment arms. We have expanded the discussion around the imbalance in *BRCA* mutation status in the discussion (lines 310–320), while taking care not to make any claims about comparisons between the two treatment arms, in line with other reviewer comments.

5. While the authors mention independent surgical review as a strength, it is unclear whether this in fact provided any added value, because apparently the blinded centralized review was based on operative notes (by the primary surgeon) rather than a second surgeon performing an independent assessment in the OR.

Response: We agree that the added value of this evaluation is not particularly clear; however, the review of change in PCI and change between CC0 and CC1 or CC2 evaluation at least allows us to have a more homogeneous analysis. This is particularly true when we see surgical reports claiming CC0 surgery although some of the abdominal cavity was not explored, or claiming no residual disease despite being unable to completely resect some peritoneal tumor lesions. We are working on a separate publication in parallel dedicated to reporting all the changes and discrepancies observed during this central review.

Reviewer #3 - Biostatistics, clinical trial design (also reviewed NCOMMS-23-08444-T)
(Remarks to the Author):

Abstract lines 36-40. These lines should be combined to conserve on word limit for other aspects of the abstract. Also, it is not clear from the description of the trial that the two arms are not being formally compared with this trial design. The trial is addressing the question whether carboplatin + paclitaxel + Pembrolizumab approaches has a complete resection rate of at least 70% as well as whether CCR with carboplatin + paclitaxel is consistent with historical findings.

Example: An open-label randomized phase II trial (NCT03275506) of women with FIGO stage IIIC/IV high-grade serous ovarian carcinoma (HGSC) for whom complete resection was unachievable was conducted to examine the complete resection rate (CCR) with carboplatin + paclitaxel and to assess whether adding Pembrolizumab to this combination yields a CCR of at least 70% (NCT03275506). Post-operatively patients continue assigned treatment for maximum of 2 years. BEV is optional.

Response: We have adapted the abstract as suggested, with some minor modifications to ensure that CONSORT-required information is retained.

Abstract line 39: Clearly state what the standard of care chemotherapy - carboplatin + paclitaxel.

Response: Adapted as suggested.

Abstract line 47-48: The presentation of pre and post operative severe toxicities leave the impression that there is a difference between treatment arms in the both the pre and post operative severe toxicity rates. It is stated that there were more post-operative toxicities with PEM than pre-operatively without a formal testing. This could be quite misleading given the small sample size and not knowing at this point in the manuscript how many patients failed to initiate post operative chemotherapy or how many patients elected to receive bevacizumab. It would be more informative to provide the most common severe toxicities seen in each treatment arm and the number of patients who discontinued due to adverse events. Post surgical complications should also be presented.

Response: The description of safety has been modified as suggested, and now includes a statement that the most common grade ≥ 3 adverse events were anemia during neoadjuvant

therapy and infection/fever postoperatively. We also mention the percentage of patients discontinuing pembrolizumab because of adverse events.

Abstract line 45 states that progression-free survival is similar between the groups. When does the time clock start for PFS at time of registration? Post surgery? It is not clear how mature these data are. What is the median length of followup? How many progressions and deaths have been seen? This should be removed from the abstract because lack clarity as to what this is.

Response: Removed from the abstract as requested. All of the requested information is included in the main text of the article.

Abstract line 50. Given your study design, a comment should be made concerning both treatment approaches, not just commenting on PEM containing regimen. Was the Carboplatin + paclitaxel similar to historical findings. Also, it is unclear why it is suggested that this be further studies in specific subgroups.

Response: In the discussion (lines 215–223, 240–244, and 323–327), we state that the carboplatin + paclitaxel arm performed much better than expected. The assumed CRR based on historical data was 50%, whereas the CRR was 70% in the standard-of-care arm. Given the limited word count, it is hard to include this level of detail in the abstract. The high-level comment on subgroups relates to subgroups defined by high PD-L1 expression, although there is not space to go into details in the short abstract. The reference to specific subgroups refers to the work in the companion paper.

Results Line 106-107. The treatment arms should not be reference to as experimental and control arms as this trial was not designed to compare treatment arms. It would be more appropriate to refer to them as investigational and standard of care (SOC) arms or NACT+/- PEM as you do in the table and figures.

Response: Adapted as requested.

Results line 113-116 The median time from randomization to IDS was 3.3 months (range 2.6–7.4 months) in the control arm and 3.2 months (range 2.4–7.3 months) in the experimental arm. This should be replaced by the time on treatment or the median number (range) of treatment cycles administered, number of dose reductions, most common reasons for dose reductions, and the reasons for discontinuation of neo-adjuvant treatment. There should be an explanation as to why 58 of 61 NACT +PEM patients and 29/30 NACT patients went on to debulking surgery. How many patients refused to undergo post operative chemotherapy? What were the results for discontinuing post-operative chemotherapy? It is important to understand how patients fared during entire treatment course not just the results of the treatment. This leads to the suggestion that toxicity section be placed before efficacy.

Response: The text cited in this comment appears within the subsection 'Patient characteristics and IDS'. We do not agree that information on the delay to IDS should be replaced with treatment exposure information. Table 2 includes all of the requested treatment exposure information (median [range] no. of cycles, dose reductions or omissions, premature discontinuations for toxicity, progression, or death, reasons for early discontinuations, etc. The CONSORT figure (originally Figure 1, now Supplementary Figure 1) provides details of the 1 patient in the standard-of-care arm and the 3 patients in the investigational arm who did not undergo IDS. It has been expanded to include the numbers of patients who received post-operative therapy. Finally, as CRR was the primary endpoint, we believe efficacy should be reported before safety (which was a secondary endpoint).

Some of the items suggested to be in the text are in Table 2 but it is difficult to figure out what is going on. It is assumed that if a patient progresses (or refuses to continue on study) all treatment is discontinued so it does not make confusing to see different number of patients under each agent being taken off for disease progression (or refuses to continue on

study). If the patient went off treatment for an AE, it is important to know the grade of toxicity. It is strongly suggested that a statement such as dose reductions were reported in carboplatin (X, %) and paclitaxel (Y,%) in SOC arm and in carboplatin (X, %), paclitaxel (Y,%) and pem (Z, %) in investigational arm. End of neo-adjuvant treatment reasons included: completed planned number of treatment cycle, toxicity, progression, etc. This can be repeated for post operative treatment as well.

Response: A patient could discontinue one component of treatment because of adverse events but continue another until disease progression, so the numbers discontinuing each drug because of disease progression do not necessarily match. We have added the grade of adverse events leading to treatment discontinuation in the table. All of the remaining information requested is in the tables and given word count limitations we have not duplicated information from the tables in the text, preferring to report novel information in the text.

Results Lines 123-125 should be removed. Secondary response endpoints showed numerically higher activity in pembrolizumab-treated patients, although there was no formal comparison of the two treatment arms (Table 1). The confidence intervals overlap.

Response: We appreciate this feedback - a similar comment was made by reviewer 1. This text has been replaced with more neutral text: 'Table 1 shows secondary response endpoints. RECIST ORR was 72% with pembrolizumab and 60% in the standard-of-care arm.'

Results Lines 128-130 state that the median duration of follow-up 129 was 22 months (range 6.8–32.5 months), progression-free survival (PFS) events had been recorded in 54 patients (59%), and 19 patients (21%) had died. The statistical analysis section fails to define PFS or duration of follow-up. Does the clock start at registration, at completion of surgery, or initiation of post operative chemotherapy?

Response: PFS is now defined in the methods section (as the interval between randomization and date of disease progression or death, whichever occurs first). Median follow-up is also calculated from the date of randomization.

Results Line 135-136. Median PFS should have accompanying 95% confidence interval.

Response: The 95% CIs were presented in the Kaplan–Meier curves and omitted to save word count in the text, but are now duplicated in the text.

Results 139: Efficacy according to PD-L1 and *BRCA* mutation status. And Figures 3-5. This entire section is an unplanned analysis not spelled out in the protocol. *BRCA* mutation status is not even mentioned in the protocol. There is no justification for combining treatment arms other than the small number of patients with either germline or somatic *BRCA* mutants. This unplanned, underpowered subset analysis of immature outcome data should be removed from the manuscript.

Response: The NeoPembrOV trial was designed as a clinical trial dedicated to testing the feasibility of immune checkpoint blockade in the neoadjuvant setting and, more importantly, to collecting biopsies before and after treatment to identify potential biomarkers. The neoadjuvant platform provides a unique opportunity for in-depth mechanistic and biomarker studies in vivo, as highlighted and analysis of markers is fundamental to the results of this study and their interpretation. When the study was designed in 2016, PARP inhibitors were not standard of care and *BRCA* mutation testing was not feasible in routine practice outside clinical trials. Since then, *BRCA* mutation status has become one of the most important factors driving treatment decision making and patient selection in ovarian cancer and is of considerable clinical importance in current treatment practice. It is also now a well-established prognostic factor, with widely differing outcomes according to *BRCA* mutation status. There is a strong expectation in the gynecologic oncology community that outcomes according to *BRCA* mutation status will be presented (please see the comments regarding analyses according to *BRCA* mutation status from reviewers 1 and 2), as these are of high

clinical relevance, although not envisaged when the trial was designed in 2016. Therefore, we are reluctant to delete these much-requested analyses and respectfully ask to retain the figure showing PFS by *BRCA* mutation status in its improved format as suggested by Reviewer 1. Similarly, PD-L1 status is of considerable relevance for immune checkpoint inhibitors. Indeed, almost all completed or ongoing phase III trials of drugs targeting PD-(L)1 have focused on the PD-L1-positive population for the primary endpoint based on the established predictive effect of this marker in other tumor types where immune checkpoint blockade is established. Therefore we believe that it is essential to report the analyses according to *BRCA* mutation and PD-L1 status in our report and this has been strengthened by updating the analyses with longer follow-up and presenting the results for the *BRCA* mutant and wildtype subsets in a single panel, and likewise presenting the PD-L1-positive and PD-L1 negative subgroups on a single panel, as suggested by Reviewer 1, point 5. Looking at the updated OS results, we continue to consider that this information may be important for the future.

Discussion Lines 190-193 Unplanned sensitivity analysis should be removed as it is unclear how this was done.

Response: The details have been provided in the results section (lines 127–131). This analysis was conducted to explore potential reasons for the higher-than-expected CRR, about which this reviewer asked above. Therefore we are reluctant to remove this work designed specifically to address such a point.

Discussion Lines 198-200 In NeoPembrOV, we observed a numerical trend in RECIST response rate favoring the addition of pembrolizumab to chemotherapy (72% versus 60% with chemotherapy alone), but no difference in PFS at the database lock for this analysis. This statement needs to be rewritten. It was clearly stated that no comparison between treatment arms were going to be made in the protocol -- there is little to no power to detect anything but a huge difference in PFS and the confidence intervals for response overlap. There is no evidence here that addition PEM is improving outcomes.

Response: Rewritten as requested.

Discussion line 261: The phrase Despite the observed trend favoring pembrolizumab should be removed. It has not been demonstrated that adding pem improves outcomes.

Response: We have deleted this phrase. Although we did not show a major improvement, a dedicated large translational program was already planned to try to understand why there was no major difference and explore whether suitable candidates for future trials could be identified.

Results Line 153-55: Overall, there was no excess of chemotherapy dose reductions or premature chemotherapy discontinuation in the pembrolizumab arm during NACT or after ID

Response: Sorry, it is unclear what change we are asked to make here. The sentence quoted from the manuscript describes the data shown in Table 2. Please clarify.

Supplemental Table 1:

- Replace mean age and standard deviation with age categorized as 40-49, 50-59, 60-69, 70-79.

Response: We consider mean and standard deviation to be a fairly standard way to present baseline characteristics. We have not analyzed the distribution of age in age categories and believe this additional analysis will bring no value and may lead to overinterpretation of any differences in distribution in relatively small sample sizes.

- Replace mean CA125 and standard deviation with CA125 categorized as < 500 U/mL, 500-999; 1000-1499; 1500-1999, 2000+ (as high pre-CA125 levels have been shown to be associated with suboptimal reductions)

Response: Quartiles are not generally used to explore the impact of treatment on CA-125. The mean level helps us to confirm the standard population enrolled compared with

historical data

Table 3: Clearly state in the title whether these are all toxicities regardless of attribution or at least possibility related to study meds.

Response: Expanded as requested (the table reports all AEs irrespective of investigator-attributed relationship to treatment).

Reviewers' Comments:

Reviewer #1:

Remarks to the Author:

My concerns have been addressed.

Reviewer #2:

Remarks to the Author:

In my opinion, reviews' comments have been satisfactorily addressed.

Reviewer #4:

Remarks to the Author:

The authors' response to the comments and explanations were reasonable. However, the interpretation of the hypothesis on CRR did not appear appropriate. According to the design, the null hypothesis of CRR was 50% and the alternative hypothesis of CRR was 70%. Meeting the criteria for the CRR endpoint implied rejecting the null hypothesis of 50% CRR. Therefore, the correct conclusion should be that CRR was greater than 50%, rather than greater than 70%. Certain statements throughout the manuscript should be revised. For example,

1. Abstract, Line 44: Modify to read "exceeding the prespecified 50% threshold." The original statement is confusing given that the lower bound of 95% CI of 63% was not above 70%.

2. Abstract, Line 40: Revise to read "... at least 50%."

3. Discussion, Line 205: it should be "...demonstrating a CRR of $\geq 50\%$ "

Additionally, it helps to clarify that the enhanced PFS and OS with combined positive score (CPS) ≥ 10 was observed only in the pembrolizumab arm (Line 161-163).

Neoadjuvant and adjuvant pembrolizumab for advanced high-grade serous carcinoma (NeoPembrOV): A randomized phase II GINECO clinical trial [Ray-Coquard et al, NCOMMS-23-08443-T

Point-by-point response to second set of reviewer comments

Reviewer #1 (Remarks to the Author):

My concerns have been addressed.

Reviewer #2 (Remarks to the Author):

In my opinion, reviews' comments have been satisfactorily addressed.

Reviewer #4 (Remarks to the Author):

The authors' response to the comments and explanations were reasonable. However, the interpretation of the hypothesis on CRR did not appear appropriate. According to the design, the null hypothesis of CRR was 50% and the alternative hypothesis of CRR was 70%. Meeting the criteria for the CRR endpoint implied rejecting the null hypothesis of 50% CRR. Therefore, the correct conclusion should be that CRR was greater than 50%, rather than greater than 70%. Certain statements throughout the manuscript should be revised. For example,

1. Abstract, Line 44: Modify to read "exceeding the prespecified 50% threshold." The original statement is confusing given that the lower bound of 95% CI of 63% was not above 70%.

Response: Modified as requested.

2. Abstract, Line 40: Revise to read "... at least 50%."

Response: Revised as requested.

3. Discussion, Line 205: it should be "...demonstrating a CRR of $\geq 50\%$ "

Response: Revised as requested.

Additionally, it helps to clarify that the enhanced PFS and OS with combined positive score (CPS) ≥ 10 was observed only in the pembrolizumab arm (Line 161-163).

Response: Revised as requested.